# Task-Decoupled Knowledge Transfer for Cross-Modality Object Detection

**DOI:** 10.3390/e25081166

**Published:** 2023-08-04

**Authors:** Chiheng Wei, Lianfa Bai, Xiaoyu Chen, Jing Han

**Affiliations:** The School of Electronic and Optical Engineering, Nanjing University of Science and Technology, Nanjing 210094, China; 319104003707@njust.edu.cn (C.W.); blf@njust.edu.cn (L.B.)

**Keywords:** cross-modality, knowledge transfer, task-decoupled pre-training, task-relevant hyperparameter evolution

## Abstract

In harsh weather conditions, the infrared modality can supplement or even replace the visible modality. However, the lack of a large-scale dataset for infrared features hinders the generation of a robust pre-training model. Most existing infrared object-detection algorithms rely on pre-training models from the visible modality, which can accelerate network convergence but also limit performance due to modality differences. In order to provide more reliable feature representation for cross-modality object detection and enhance its performance, this paper investigates the impact of various task-relevant features on cross-modality object detection and proposes a knowledge transfer algorithm based on classification and localization decoupling analysis. A task-decoupled pre-training method is introduced to adjust the attributes of various tasks learned by the pre-training model. For the training phase, a task-relevant hyperparameter evolution method is proposed to increase the network’s adaptability to attribute changes in pre-training weights. Our proposed method improves the accuracy of multiple modalities in multiple datasets, with experimental results on the FLIR ADAS dataset reaching a state-of-the-art level and surpassing most multi-spectral object-detection methods.

## 1. Introduction

The advancement of camera sensors has improved information-collecting channels and made it feasible to gather data from several cameras with various spectra. The use of visible and infrared modalities in downstream computer vision applications has increased recently, particularly in the area of object detection. In dealing with complicated occasions, infrared data based on the thermal radiation properties of the object has more advantages and can be extremely useful. For example, there have been some related studies on object detection paired with infrared modalities in the field of autonomous driving, as well as open-source object-detection datasets incorporating infrared modality data [1,2]. However, the volume of these datasets is far smaller than the existing large-scale visible modality datasets [3,4], and there are significantly fewer open-source object-identification datasets using infrared modalities in other application fields, such as airborne. One of the challenges in the advancement of infrared object detection is the gathering of infrared data. The purpose of infrared object detection is to efficiently use the limited infrared data to mine the target’s infrared features for detection.

Researchers typically use transfer learning, image generation [5,6], and image fusion [7] to improve network performance in order to fully utilize different modality information. The transfer learning method initially performs fine-tuning in subsequent detection tasks after training a pre-training model on a large-scale visible classification dataset. Such a technique can dramatically increase detection accuracy while speeding up network convergence on a limited dataset. In order to increase the total number of infrared data in the target domain and enhance the precision of infrared object detection, some researchers use image-generation techniques to transform visible image data into pseudo-infrared images [8]. In addition, image fusion methods are used by researchers to fuse visible and infrared modalities in the network [9,10,11]. The network can learn better target representation and improve the subsequent detection effect by incorporating infrared modality information based on visible object detection. However, due to the spectral characteristic differences between the visible and infrared modalities, it is difficult for the image-generation method to provide reliable infrared image features, which limits network performance. Image fusion based on multi-spectral images needs to learn effective feature representations based on data; however, limited by the scale of training data, it is difficult for the network to obtain robust generalization capabilities on small-batch datasets.

We research cross-modality object detection using transfer learning methods to investigate how to efficiently use limited data. The visible pre-training model is used to initialize the majority of existing infrared object-detection algorithms. However, there are still two barriers to establishing a good transfer effect: task-level differences and modality-level differences. Recent research has revealed that ImageNet-based pre-training does not improve object-detection accuracy on the COCO dataset [12]. When object classification is the upstream task and object detection is the downstream task, the feature learned by the pre-training model will overfit the upstream classification task, resulting in a lack of localization-related information required by the downstream detection task and limiting the infrared detector’s performance. Since the network’s weight initialization has a significant impact on the training process, the modality-level difference is mainly reflected in the difference between visible imaging and infrared imaging. Visible imaging mainly reflects the reflection characteristics of the target, while infrared imaging reflects the radiation characteristics of the target. The difference in imaging mechanism results in a difference in target characteristics between visible and infrared modalities, which is primarily reflected in the target’s details, texture, and color brightness, while the difference in edge contour and corner information related to the target’s scale and shape is minor. These changes also affect detector responses to distinct target features, as well as the activation of associated convolution kernels in different modalities.

To achieve cross-modality information transfer, we use heatmaps to visualize and compare the final output features of the visible and infrared modality network models, including localization regression and classification features. The heatmap is a graphical representation of the responsiveness of image features within a neural network. Figure 1 depicts the original image and related heatmaps. From the comparison diagram in the last row, it can be clearly seen that the differences in the object texture and other information of different modalities in the original image and the heatmaps related to object classification are quite different, as the spatial heatmap responses of pedestrians and bicycles are different, while the heatmaps related to localization are consistent. In other words, object localization is less influenced since it pays more attention to regression-sensitive feature information such as the object contour, whereas object classification is highly impacted by the classification-sensitive feature variations in the data of different modalities. In the field of object detection and remote sensing, some work has been optimized and improved around the classification-sensitive feature and the regression-sensitive feature [13,14,15].

This finding motivates us to reconsider the current visible pre-training approaches. If the pre-training model can be modulated, would the network be able to extract additional useful cross-modality features from the pre-training model? We propose a task-decoupled pre-training (TDP) method to lower the transfer barrier of cross-modality features based on the aforementioned concepts. By decoupling the classification and localization attributes in the pre-trained model, the TDP method aims to improve the learning of localization-related features that are insensitive to modality changes during the training process, thereby improving the transfer effect of the pre-training model in subsequent target domain tasks. In addition, we propose a task-related hyperparameter evolution (TRHE) technique to improve the network’s TDP accuracy even further. The TRHE method focuses on the flexibility of downstream tasks to the modulated pre-training model, in contrast to standard hyperparameter optimization methods. To prevent TDP from introducing negative transfer into the downstream training process, this research employs evolutionary algorithms for classification-related and localization-related hyperparameters to improve the training process’s adaptability to varied pre-training models. Our methods are only applicable during the training phase; the inference phase is not adjusted compared to the baseline method, and these methods can be easily merged with other cross-modality object-detection methods.

Our contributions are summarized as follows:We rethink the generally used visible classification pre-training approaches and propose the TDP method by decoupling classification and localization features to obtain a pre-training model that is more conducive to cross-modalities.Further, we propose the TRHE method to adjust the hyperparameters related to classification and localization during the training process and improve the adaptability of the network to the modulated pre-training model.We investigate the influence of modality changes on the detection network’s classification and localization components and validate the effectiveness of our methods on MSOD and FLIR datasets while achieving state-of-the-art accuracy on the FLIR dataset, surpassing most multi-spectral object-detection benchmarks.

## 2. Related Works

### 2.1. Multi-Spectral Object Detection

Object detection has advanced considerably in recent years, with visible object detection achieving great success [16,17,18,19,20]. Simultaneously, infrared object-detection technology is fast evolving in order to enable object detection at night. Ghose et al. [21] use significance detection to extract significant features from infrared data, guide the feature-learning process, and perform well on the KAIST pedestrian detection dataset [22]. Cao et al. [23] refined the feature fusion method that was employed on the FLIR infrared modality, which was based on the RefineDet network. In comparison to the typical visible modality, infrared can meet detection requirements in scenes such as nighttime and overexposure that the visible modality cannot. However, due to the limited number of data, the detector cannot meet the needs of object detection in some complex weather conditions when relying on single-spectral images. Multi-spectral object detection brings improvements by introducing image information of objects in a multi-spectral format. The detector benefits from the rich color information of the visible modality and may use the infrared modality information to suit the needs of object detection at night after merging the visible modality and long-wave infrared modality. Liu et al. [24] investigated various fusion strategies for visible and infrared modalities using the Faster RCNN architecture. GFD-SSD [25] adopts the Gated Fusion module to fuse the two modality branches. Zhang et al. [26] focused on the weak alignment problem between visible and infrared modality data, which is common in similar datasets due to cameras or alignment algorithms. Zhou et al. [27] introduced the modality-imbalance problem between two modality data and proposed the Differential-Modality-Aware Fusion module to solve the modality imbalance problem and realize the fusion of different modality data.

However, there is a challenge in both non-visible modality single-spectral object detection and multi-spectral object detection, that is, the difficulty of data acquisition. Non-visible modality datasets are currently substantially smaller in scale than visible modality datasets in existing public datasets, and multi-spectral object identification datasets must handle the alignment of distinct modality data, which increases the difficulty of data gathering. As a result, effective pre-training models for non-visual modalities from large-scale datasets of their own modalities, such as the visible modality, are lacking. The training process is frequently started with pre-training models of the visible modality, which limits the detector’s performance. The pre-training model is a crucial aspect of improving the performance of non-visible modality detectors.

### 2.2. Cross-Modality Knowledge Transfer Based on Pre-Training Model and Fine-Tuning

The pre-training model and fine-tuning are transfer learning methods, and they are also the most commonly used methods for cross-modality knowledge transfer. Firstly, the pre-training model is obtained through training on the large dataset. Then fine-tuning is performed on the target domain dataset, transferring the knowledge learned on the large dataset to the target domain dataset, and improving the network performance on the target domain dataset. For computer vision tasks, the ImageNet supervised pre-training method is a commonly used model initialization technique. However, the latest works show that when the distance between the source domain and the target domain is large or the target domain has sufficient training data, the supervised pre-training model does not bring improvement [12]. The supervised pre-training method concentrates on category-level discrimination, whereas the self-supervised pre-training method focuses on instance-level discrimination, which can yield more discriminative features in subsequent tasks. Zoph et al. [28] used the self-supervised pre-training method to improve object detection and other visual tasks with strong data augmentation. However, due to the lack of a supervision mechanism, the self-supervised pre-training method lacks the ability to mine high-dimensional semantic features, and it can easily obtain redundant and irrelevant features [29]. In recent years, relevant transfer learning research has focused on the transfer obstacles created by variations between upstream and downstream tasks, as well as on improving the transfer effect across tasks. This paper investigates the pre-training method of multi-spectral object detection. In order to reduce the loss of transfer effect caused by cross-tasks, we first train the pre-training model on the visible large-scale object detection dataset and then fine-tune it on the cross-modality object-detection dataset.

We observe the difference between classification-sensitive features and regression-sensitive features. Classification pays more attention to classification-sensitive feature information such as the texture and color of the object, while regression pays more attention to regression-sensitive feature information such as the outline and corner points of the object. Existing transfer learning approaches for downstream object detection ignore the distinction between the two types of information. Regression-sensitive features, such as contours and corners, can be better retained in source- and target-domain data than classification-sensitive features, such as color and brightness. Therefore, in order to allow the pre-training model to learn more effective features in the target domain, we propose the task-decoupled pre-training (TDP) method to modulate the pre-training model during training in order to retain more regression-sensitive features of the source domain and thus improve the pre-training method’s effectiveness. Compared with previous work, the new method is optimized based on classification-sensitive features and regression-sensitive features, which is more direct and effective than self-supervised pre-training methods.

### 2.3. Hyperparameter Optimization

Hyperparameter optimization (HPO) is a well-known black-box optimization issue. With the increasing scale of neural networks and the increasing number of hyperparameters, HPO methods have become popular. The overall concept of HPO is to first determine the hyperparameters that need to be searched, as well as the search range, and then conduct iterative training to evaluate the performance of the model after the combination of different hyperparameters, and finally, to find the most effective combination [30]. In order to improve the accuracy of object detection, researchers often consider optimizing hyperparameters to improve the effectiveness of the training process. Ma et al. [31] used the HPO method to optimize the setting parameters of the anchor box of the detector and obtained a robust detection effect. In order to improve the effect of pedestrian detection, Gagneja et al. [32] optimized some hyperparameters of FPN based on the HPO method and studied the influence of different network parameters. The above work employed HPO to improve network adaptation to datasets and achieved promising results. However, HPO approaches are infrequently applied in existing work to increase the generalization performance of cross-domain object-detection tasks.

The subsequent training process of the network is unstable in cross-modality detection tasks that are affected by the modality difference between the upstream and downstream tasks. As a result, the hyperparameters acquired by typical HPO methods after a lengthy search are not acceptable sub-optimal solutions. To address this issue, we modify the existing HPO method and propose the TRHE method, which is based on the existing hyperparameter evolution method. The major goal of the TRHE approach is to enhance the network’s adaptability to the modulated pre-training model while also reducing the time required for hyperparameter evolution in order to obtain an acceptable sub-optimal solution under the aforementioned premise. The goal is achieved by only searching for task-related hyperparameters and reducing the number of hyperparameter search iterations.

## 3. Method

The cross-domain process uses the features learned in the visible source domain, such as infrared, in the target domain to improve the accuracy of the network in the target domain. To effectively exploit the visible pre-training model and reduce the model’s accuracy limitation, we proposed the task-decoupled pre-training (TDP) method and the task-related hyperparameter evolution (TRHE) method. The main goal of the method proposed in this paper is to make hyperparameter adjustments to the network’s pre-training model and training process based on the concept of feature decoupling, which primarily adjusts the gain values related to the classification task and the regression task in the loss function so that the network can obtain more effective cross-domain features from the pre-training model and improve the network’s adaptability to feature changes. The algorithm flow is shown in Figure 2. The final model is trained on the basis of the pre-training model obtained using the TDP method and the hyperparameters obtained using the TRHE method.

### 3.1. Overview

Firstly, we will describe the method and some definitions of the network training process. We set the data of the source domain and target domain to $D_{S}$ and $D_{T}$, the default hyperparameters and pre-training model to $\theta_{D}$ and $P_{D}$, the hyperparameters adjusted using the TDP method, and the pre-training model obtained using the TDP method to $\theta_{TDP}$ and $P_{TDP}$, the hyperparameters initialized of the TRHE method and obtained using the TRHE method to $\theta_{I}$ and $\theta_{TRHE}$. The object-detection network used and the corresponding loss function are Ψ and *L*, respectively, and the initialized model weight is $\omega_{init}$. The task-related hyperparameter evolution function is *T*, and the fitness function used in the evolution process is *f*. The general object-detection training process seeks to minimize the value of L(ω):(1)$$\begin{array}{cc} \; & \min_{\omega }\;L\left(\omega \right)=\left\{\frac{1}{n}\sum_{i=1}^{n}L(\Psi \left(x_{i},\theta_{D},\omega_{init}\right),y_{i})\right\},\\ & \;\;\;s.t.\;\omega_{init}=P_{D},x_{i}\in D_{T} \end{array}$$

In Equation (Equation 1), $L(\Psi \left(x_{i},\theta_{D},\omega_{init}\right),y_{i})$ is the loss after each iteration, where $x_{i}$ and $y_{i}$ are the training samples and labels, respectively; Ψ and $\theta_{D}$ denote the detection network and default hyperparameters mentioned above; and the model uses the default pre-training model $P_{D}$ to initialize the weight $\omega_{init}$. The training data come from the target domain data $D_{T}$, and the pre-training model $P_{D}$ is trained on a large visible dataset from the source domain based on the default hyperparameters. The final training process of our method is the same as the general training process, but the pre-training model and training hyperparameters are obtained using the TDP method and the TRHE method, respectively. The three stages of the process, the TDP method, the TRHE method, and the final training, are shown in Figure 2, and our final training process is shown in Equation (Equation 2). Some of the notations used in this equation are consistent with those in Equation (Equation 1). However, Equation (Equation 2) uses the pre-training model $P_{TDP}$ obtained using the TDP method, the hyperparameter $\theta_{TRHE}$ obtained using the TRHE method, and the target domain data $D_{T}$. After the pre-training model and task-related hyperparameters are adjusted in the training process, the network model can obtain a more robust performance.
(2)$$\begin{array}{cc} \; & \min_{\omega }\;L\left(\omega \right)=\left\{\frac{1}{n}\sum_{i=1}^{n}L(\Psi \left(x_{i},\theta_{TRHE},\omega_{init}\right),y_{i})\right\},\\ & \;\;\;\;s.t.\;\omega_{init}=P_{TDP},x_{i}\in D_{T} \end{array}$$

### 3.2. Task-Decoupled Pre-Training

The typical ImageNet-based pre-training method would fail due to large domain gaps in downstream object-detection tasks, particularly in multi-spectral object-detection tasks, where variances across different modality objects raise the domain gap even further. Despite the fact that the pre-training model was trained on the COCO dataset, there are still differences in cross-modality characteristics. To reduce the impact of feature differences between modalities, we adjust the gain values $G_{C}$ and $G_{R}$ of the classification part and regression part in the training loss function based on the task decoupling idea so that the pre-training model can learn more regression-sensitive features, thereby improving the transfer ability of the pre-training model. Typically, the network loss function consists of three parts, namely, the classification loss $L_{cls}$ used to constrain the anchor category, the regression loss $L_{reg}$ used to constrain the anchor position localization, and the objectness loss $L_{obj}$ used to constrain the anchor confidence. The loss function is modulated using the TDP method as shown below:(3)$$L=G_{O}\times L_{obj}+G_{C}\times L_{cls}+G_{R}\times L_{reg}$$
(4)$$\begin{array}{cc} \; & \min_{\omega }\;L_{TDP}\left(\omega \right)=\left\{\frac{1}{n}\sum_{i=1}^{n}L(\Psi \left(x_{i},\theta_{TDP},\omega_{init}\right),y_{i})\right\},\\ & \;\;\;\;s.t.\;\omega_{init}=\omega_{rand},x_{i}\in D_{S} \end{array}$$

After the TDP method is used, the loss function is formed of classification loss, regression loss, and objectness loss multiplied by the associated gain, as shown in Equation (Equation 3). During the training of the pre-training model, we increase the weight of the regression part in the loss function and then improve the regression-sensitive features learned by the pre-training model. Equation (Equation 4) depicts the training procedure. $P_{TDP}$ is trained on the source domain dataset $D_{S}$ using the hyperparameter $\theta_{TDP}$ adjusted by the TDP method. Since the pre-training model is trained on the large-scale visible dataset COCO, the training process directly adopts the parameter random initialization strategy to train from scratch. Experiments show that the modulated pre-training model can greatly improve the network’s detection accuracy on the target domain.

### 3.3. Task-Related Hyperparameter Evolution

Hyperparameter evolution is a kind of hyperparameter-optimization method, which is based on a genetic algorithm for hyperparameter mutation. The parent sample selection method includes the random selection and best-sample selection, and the mutation is realized depending on the parent sample. It should be noted that the best-sample-selection method only selects a single sample as the parent sample to increase the sampling rate of the algorithm around the best sample. After obtaining the mutated hyperparameters, the network is trained based on the pre-training model $P_{TDP}$, and the fitness function *f* is utilized to evaluate the results of multiple iterations and steer the next optimization direction. The fitness function is constructed based on the object-detection evaluation metric mAP, and the definitions of mAP and fitness function are as follows:(5)$$mAP^{IoU}=\frac{1}{C}\sum_{j}^{C}AP_{j}^{IoU}$$
(6)$$f=\lambda_{1}\times mAP+\lambda_{2}\times mAP50$$

In Equation (Equation 5), the mAP of different Intersections over Union (IoUs) is obtained by calculating the mean AP [33] value of each type of object under different IoU conditions, and *C* is the number of categories. The mAP in this paper is the evaluation metric of the COCO object detection dataset [3], and the mAP50 is calculated when the IoU is 0.5. Equation (Equation 6) is the definition of the fitness function *f*. In order to balance the accuracy of mAP and mAP50, the fitness function multiplies the weights of the two metrics with $\lambda_{1}$ and $\lambda_{2}$ and sums them up. Experience shows that when $\lambda_{1}$ is 0.9 and $\lambda_{2}$ is 0.1, hyperparameter evolution works better [34]. The mutation process of the parent sample is shown in Equation (Equation 7), where *p* is the mutation probability of a single hyperparameter in the parent sample; *v* is the magnitude of the mutation; $r_{1}$ and $r_{2}$ are the random numbers participating in the mutation, which follow the normal distribution and uniform distribution, respectively; $\theta_{i-1}^{best}$ is the hyperparameter with the best fitness in the previous iteration process; and $\theta_{i}^{mut}$ is the mutation result of the hyperparameter in this iteration.
(7)$$\begin{array}{cc} \; & \theta_{i}^{mut}=\theta_{i-1}^{best}\times \left(1+p\times r_{1}\times r_{2}\times v\right),\\ & \;s.t.\;r_{1}∼N(0,1),r_{2}∼U(0,1) \end{array}$$
(8)$$\theta_{TRHE}=T\left(\theta^{mut}\right),\;s.t.\;G_{C},G_{R}\in \theta^{mut}$$
(9)$$\begin{array}{cc} \; & T\left(\theta \right)=\arg\max_{\theta }f(\Psi \left(x,\theta ,\omega_{init}\right)),\\ & \;s.t.\;\omega_{init}=P_{TDP},x\in D_{S} \end{array}$$

During the experiment, we set the mutation probability *p* to 0.8 and the variation magnitude *v* to 0.04 according to experience [34]. Equation (Equation 8) depicts the formula for task-related hyperparameter evolution. The parent sample is chosen based on the fitness function *f* in each iteration process, and the parameters of the following iteration are acquired using the mutation algorithm. After multiple iterations of training, the final hyperparameter $\theta_{TRHE}$ is obtained. The formula for a single iteration is shown in Equation (Equation 9). By optimizing the hyperparameters based on $P_{TDP}$ on the small dataset of the infrared modality in the target domain, compared with using the default hyperparameter $\theta_{D}$, the training process improves the adaptability of the network to the pre-training model after modulation, which is further improved on the basis of the TDP method. Meanwhile, due to the modality difference between the knowledge distribution in the pre-training model and the target domain’s data distribution, the training process is unstable, which may result in an unacceptable suboptimal solution. Experiments have found that the randomness of the $G_{C}$ and $G_{R}$, the gain values of classification, and the regression loss in the loss function can easily lead to a decrease in the final accuracy. In order to obtain a more stable result, we further propose a binding mode of TRHE to maintain the ratio of classification and regression loss gains with the value α, as shown in Equation (Equation 10).
(10)$$G_{R}=\alpha G_{C}$$

The experimental results show that more stable results can be obtained by locking the ratio of classification and regression gains during the evolution of task-related hyperparameters.

### 3.4. The Algorithm Combining TDP and TRHE Methods

In order to show our proposed algorithm more intuitively, Algorithm 1 shows the whole process of combining the TDP and TRHE methods.

The three main processes described in each line are as follows:Line 1: Modulate the training process of the pre-training model based on the TDP method, and the source domain data come from a large visible detection dataset.Lines 2–15: The TRHE method is used to obtain effective hyperparameters, and the training is performed on the target domain dataset using the pre-training model obtained in the previous process. In the method with N iterations, the initial hyperparameters are changed using the genetic mutation algorithm and evaluated using the fitness functions. Finally, better hyperparameters are produced.Line 16: The final detector is trained on the target domain dataset using the modulated pre-training model and hyperparameters obtained in the previous two processes.
**Algorithm 1:** Complete algorithm with TDP and TRHE.
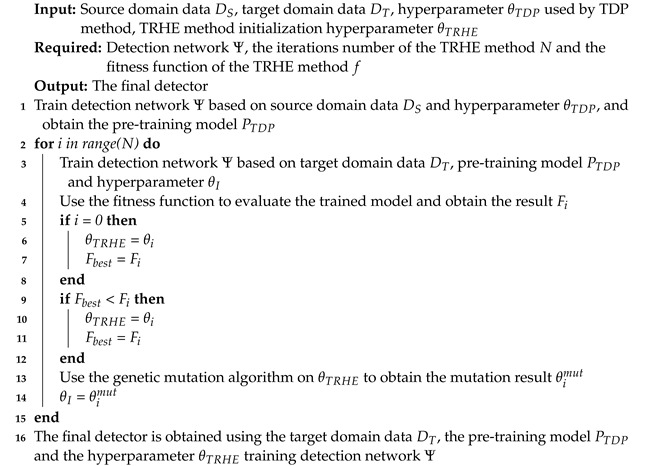


## 4. Experiments

In this section, we first verify the effectiveness of the task-decoupled pre-training (TDP) method on the MSOD dataset and the FLIR dataset, then explore the influence of the pre-training model after the modulation of different hyperparameters on the training results of subsequent detection tasks, and further verify the effectiveness of the task-related hyperparameter evolution (TRHE) method.

### 4.1. Experimental Configuration

#### 4.1.1. Datasets

Multi-spectral Object Detection Dataset. MSOD dataset [2] comprises image data of four different modalities, namely visible, near-infrared, mid-wave infrared, and long-wave infrared. The dataset contains 7521 unaligned image pairs and 2999 aligned image pairs. Visible and long-wave infrared images have a resolution of 640×480, while near-infrared and mid-wave infrared images have a resolution of 320×256. Three common objects, person, car, and bike, were used in the experiment. In our experiments, we used the MSOD dataset to test the transferability of our method on multiple modality data. To investigate the impact of our method on multi-spectral object detection when the training and test sets are significantly different, we selected 738 pairs of images from all unaligned data according to different scenes during the day and night as the testing set, and the remaining 6783 pairs of images were used as the training set.

FLIR Dataset. The FLIR dataset [1] comprises two modalities of images: visible with a resolution of 1280×1024 and long-wave infrared with a resolution of 640×512. The researchers selected the original FLIR dataset and rearranged it into a multi-spectral object-detection dataset with good alignment [35]. The new alignment dataset contains 5142 pairs of well-aligned images, with 4129 pairs in the training set and 1013 pairs in the testing set. The FLIR-aligned dataset contains three types of objects, namely pedestrian, car, and bicycle. In our experiments, we used the FLIR-aligned dataset to evaluate our method on infrared and mixed-modality data.

#### 4.1.2. Implementation Details

We conducted various experiments using the two scale networks, YOLOm and YOLOx, within the YOLOv5 framework [34], to test the performance of our method under medium- and large-scale backbone networks. We used the default configuration for training the COCO dataset as the hyperparameter initialization for training. The network was optimized using SGD optimization, with an initial learning rate of 0.01, a momentum value of 0.937, and a weight decay of 0.0005. To mitigate the impact of network convergence randomness on our experiment, we averaged the indicators of three repeated instances of training to obtain the results presented in this paper.

In this paper, we obtained the pre-training model by training on the COCO dataset. Due to the large size of the COCO dataset, we chose to train from scratch when preparing the pre-training model. All pre-training models trained 30 epochs on the COCO dataset rather than training the network to complete convergence. On the one hand, we discovered through simple tests that the incompletely converged pre-training model can achieve or even exceed the transferability of the fully converged pre-training model on cross-modality datasets. On the other hand, when the hyperparameters are adjusted, the network cannot converge entirely under specific parameter-setting conditions.

In our experiment, the regN model represents the pre-training model trained with $G_{R}$ of N and $G_{C}$ of 0.5, while the clsN model represents the pre-training model trained with $G_{R}$ of 0.05 and $G_{C}$ of N. More experimental details of the guided hyperparameter sweep and the TRHE method are presented in the following ablation experiments.

### 4.2. Main Results

To verify the effectiveness of our method, we compared it with several existing state-of-the-art methods, as described in this subsection, using the FLIR dataset for all experimental results. These methods mainly include image fusion methods [35,36,37,38,39], image enhancement [23], and image generation [8]. From Table 1, we can draw several conclusions: (1) The performance of the two detection networks based on image-enhancement or image-generation methods is slightly inferior to that of the image fusion method. However, our method can achieve the accuracy level of the image fusion method while only using infrared modality data during the test phase. (2) The optimization of the pre-training model allows our method to improve even when using a large backbone network. YOLOx further improved by 1.57% compared to YOLOm. (3) Compared with the ProbEn method, which achieves the highest accuracy in the multi-spectral method, our method is slightly worse in the category of pedestrians and better in the category of cars. The results based on yolox can achieve more than 0.64% improvement on mAP50.

Image fusion methods require multi-spectral data during both the training and testing phases. With effective fusion strategies, these methods can fully leverage the diverse properties of multiple spectra. ProbEn is based on effective late fusion in the above manner to achieve the highest accuracy among current fusion methods. However, effective fusion frequently requires the extraction of effective features in the latter half of the network to avoid the loss of multiple types of modality information, which leads to a huge network volume and a long inference time. In contrast, transfer learning methods only require single-spectral data during the testing phase, reducing the need for multi-modality data. Despite the lack of multi-spectral feature information in the inference stage network, the network’s performance on a single modality can be mined using pre-training methods or image-enhancement and image-generation methods, and a state-of-the-art level of accuracy can also be obtained. Our method outperforms most image-fusion methods while achieving the best result among single-spectral methods. The approach achieves the highest accuracy using the yolox model.

### 4.3. Ablation Study

As described in the implementation details in Section 4.1.2, in this section, we conducted a series of ablation experiments on the TDP and TRHE methods to investigate their effects on the FLIR and MSOD datasets.

#### 4.3.1. Results and Discussion of the TDP Method

We conducted experiments on the MSOD dataset’s multiple modalities, along with the FLIR dataset’s infrared modality and mixed modality, to investigate the effectiveness of the TDP method. Table 2 shows the results of fine-tuning based on the pre-training model constrained by different regression attributes on the MSOD long-wave infrared modality. Table 3 presents the results of fine-tuning based on the pre-training model constrained by different classification attributes. We also conducted tests on the near-infrared and medium-wave infrared modalities of the MSOD dataset, with the experimental results shown in Figure 3.

On the MSOD long-wave infrared dataset, increasing the regression attribute constraints of the pre-training model results in a 3.3% improvement over the pre-training model using the default hyperparameters. Similarly, on both the near-infrared and mid-wave infrared modalities of the MSOD dataset, the pre-training model with a larger regression gain constraint improves by 1.39% on the near-infrared modality and 1.26% on the mid-wave modality compared to the pre-training model with the default hyperparameters. It should be noted that increasing the classification gain constraint on the long-wave infrared modality may result in a significant decrease in accuracy while improving accuracy on the near-infrared and medium-wave infrared modalities. We believe that this is due to the higher classification-sensitive feature similarity between the near-infrared and mid-wave infrared modalities and the visible modality.

On the FLIR dataset, we also investigate the effects of the TDP method on long-wave infrared and mixed modality. We performed channel-level interaction directly on visible and long-wave infrared data to obtain mixed modality data for testing. First, we used the default pre-training model to test different mixing methods. Since the annotations of the FLIR dataset are based on infrared images, there is a misalignment issue between visible features and labels. As a result, the more visible features in the mixed image, the lower the final accuracy. The experimental results are shown in Table 4.

To achieve the best results, we performed the mixed-modality experiment on the TTB modality, which had the highest accuracy. The experimental results of two groups of different gain value constraints on the infrared modality are shown in Table 5 and Table 6. Figure 4 shows the effect of different pre-training models on the infrared and mixed-modality data. On the FLIR infrared dataset, increasing the regression attribute constraints of the pre-training model resulted in a 3.03% improvement compared to the pre-training model using the default hyperparameters. The mixed modality also showed a 2.22% improvement.

The experimental results on the two datasets confirm the effectiveness of the TDP method. Regression attribute features may have a better transfer effect than classification attribute features in cross-modality transfer learning. The reasonable addition of regression attribute features in the pre-training model can improve the transfer effect and improve the accuracy of the network in the target domain. However, the experiment also demonstrates that in order to obtain the best generalization impact on different datasets, the modulation parameters of the classification and regression attributes in the pre-training model must be variable, and fixed modulation values cannot adapt to any training set. Therefore, we introduced the TRHE method on top of the TDP method during network training to search for hyperparameters related to classification and regression tasks, thereby improving the network’s adaptability during the training phase.

#### 4.3.2. Comparison of Different Pre-Training Models and Hyperparameter Training Results

Before delving into the TRHE method, we investigate the impact of hyperparameters on final training results on the MSOD long-wave infrared dataset using the hyperparameter sweep method under different attributes’ pre-training model conditions. The experiment employed two groups of comparative experiments, which were carried out on two example pre-training models, the Reg0.7 model, with a $G_{R}$ of 0.7 and a $G_{C}$ of 0.5, and the Cls1.0 model, with a $G_{R}$ of 0.05 and a $G_{C}$ of 1.0. The sweep method simply modifies the loss classification and regression gain values in the hyperparameters, leaving the other hyperparameters unchanged. The search method is based on a random search, with each iteration training 10 epochs and each group of experiments iterating 100 times. The visualization results of the guided hyperparameter sweep are shown in Figure 5. The picture depicts many hyperparameters, as well as the matching accuracy chromaticity bars. The darker the hue of the curve is, the less effective the hyperparameters are. The optimal classification gain value and regression gain value of the Reg0.7 pre-training model are 0.165 and 0.0215, and the optimal classification gain value and regression gain value of the Cls1.0 pre-training model are 0.2566 and 0.02875.

The figure shows that when the pre-training model has more classification-sensitive features in the left column, the network is more adaptable to a larger regression gain value, and a larger classification gain value results in a darker color of the upper half of the curve in the second left figure and poor training results. The network is more adaptable to the large classification gain value when the right column’s pre-training model contains more regression-sensitive features, and the larger regression gain value results in darker curves in the upper half of the first right figure and inferior training results. This demonstrates that when there are more regression-sensitive features in the pre-training model, a bigger regression gain value decreases accuracy; when there are more classification-sensitive features in the pre-training model, a larger classification gain value decreases accuracy. Simultaneously, the more effective range of the hyperparameter search may be determined: the effective classification gain value should be less than 1.0, and the effective regression gain value should be less than 0.1. Experiments have verified the sensitivity of network training results to changes in task-related hyperparameters after fixing the pre-training model and task-independent hyperparameters, which further reflects the importance of the TRHE method.

#### 4.3.3. Results and Discussion of the TRHE Method

We employed the TRHE method to adapt the network training process to changes in the pre-training model’s characteristics, resulting in greater accuracy. The experiment was carried out on the long-wave infrared modality of the FLIR dataset. In order to explore the effect of hyperparameter evolution in more conventional scenarios, we randomly chose the pre-training model Reg0.5 with greater regression attribute characteristics to conduct the TRHE experiments. The initial value of the hyperparameter had a significant impact on the effect of the TRHE method. We experimented with different initial values of the hyperparameter. The TRHE algorithm’s parent-sample-selection methods include random selection and best-sample selection; we also conducted related experiments on different selection methods. In the experiments, the α of the binding TRHE method is set to 0.1, which is consistent with the initial value ratio. Each training iteration trains 25 epochs, and each group of experiments iterates 20 times. It should be mentioned that the number of iterations and training times change depending on the goal of the experiment. The guided hyperparameter sweep is to explore the variations in the training process of different pre-training models. Observing the trend necessitates the use of multiple sets of hyperparameters. The goal of the TRHE approach is to produce more effective hyperparameters than the initial hyperparameters, which necessitates more in-depth training and evaluation of each set of hyperparameters. The experimental results are shown in Table 7. Three groups of different initial values are each compared using four methods, namely, baseline, TRHE method with random parent sample selection, TRHE method with best parent sample selection, and the binding TRHE method with best parent sample selection (TRHE*).

The experimental results show that when the initial values of the classification gain $G_{C}$ and regression gain $G_{R}$ are 0.5 and 0.05, respectively; the adaptability of the TRHE method is higher; and the performance of the metrics is such that the values of mAP and mAP50 are relatively high. At the same time, the experiment also reflects the randomness in the optimization of the TRHE method. The two parent-sample-selection methods of the random and best selection in the TRHE method may have the possibility of negative optimization because of the randomness in the optimization process. The hyperparameters of the proposed binding TRHE method (TRHE*) have less random variation and frequently obtain more stable optimization results, resulting in more stable network results. In the experiment, the best fitness optimization result is obtained based on TRHE*.

#### 4.3.4. Ablation Experiments on the TDP and TRHE Methods

We conducted ablation experiments on the infrared modalities of the FLIR and MSOD datasets in order to investigate the respective effects of the two methods. The experimental results are shown in Table 8. Experiments show that when the two methods are used on the two datasets, both of them can bring improvement, although the improvement brought about by the TRHE method is limited. When the TDP method is used combined with the TRHE method, the network can be further improved.

On the FLIR dataset, the improvement obtained by using only the TRHE method is minimal, while on the basis of the TDP method, the improvement brought about by using TRHE is greater. Although the TRHE method does not improve significantly on the basis of the TDP method, it can make network training accuracy more stable. When the optimal pre-training model cannot be determined, it can provide a supplement to the TDP method, allowing the network to be more adaptable to any pre-training model. At the same time, the experiment confirmed that the pre-training method has a larger impact on network accuracy than the post-tuning method, which also reflects the importance of optimizing the pre-training model in the cross-modality detection task.

### 4.4. Discussion of the Idea of Task Decoupling

Through the visualization of the classification attribute features and regression attribute features, we obtained a prior that the responsiveness of regression attribute features is similar among different modality features. On this basis, we designed a task-decoupling method to improve the performance of pre-trained models in cross-modality object-detection tasks. Thanks to the convenience of Equation (Equation 3), we can achieve the decoupling of classification and regression features by adjusting the coefficients in the equation. In extreme cases, the pre-trained model can learn only single-task attribute features (when $G_{C}$ is 0 or $G_{R}$ is 0). However, experimental results show that completely removing either classification or regression loss may lead to a decrease in the performance of the pre-training model. We hypothesize that when the coefficient of a certain task is too small or too large, it may cause an imbalance in the learning of model features, leading to a decrease in performance. Therefore, the coefficients should be adjusted within an appropriate range to obtain a better pre-training model. Furthermore, based on the prior and the expected experimental results, we believe that by adjusting the loss function of the pre-training model, the pre-training model can learn richer regression features and improve its performance in downstream cross-modality object-detection tasks.

## 5. Conclusions

In this research, we investigated the cross-modality object detection pre-training method, optimized it based on classification-sensitive and regression-sensitive features, and improved the transfer effect of the pre-training model using the task-decoupled pre-training (TDP) method. During the training process, the task-related hyperparameter evolution (TRHE) method was further used to obtain more effective hyperparameters to improve the adaptability of the training process to the pre-training model. Our method achieved considerable improvement on small single spectral datasets, particularly in the infrared modality, where it achieved state-of-the-art accuracy without increasing the burden of the inference process. However, the method still has some limitations; that is, when the number of target domain data is enough, the transfer learning method based on pre-training is less effective. In future research, we will explore more effective cross-domain pre-training methods and combine them with image fusion methods to address the limitations of our current approach.

## Figures and Tables

**Figure 1 entropy-25-01166-f001:**
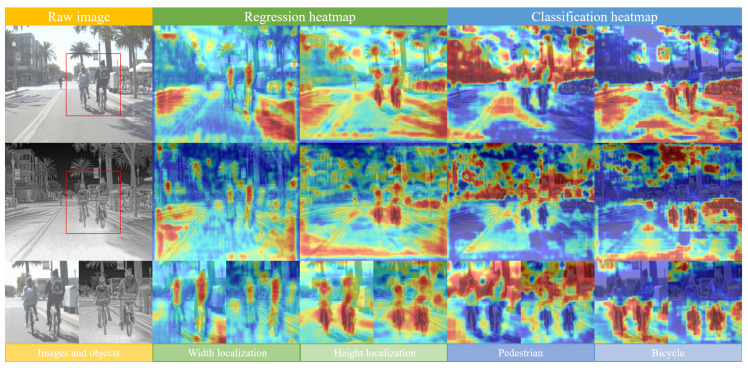
Visualization result of multi-modality classification and localization features from the final output features of the network. The first row displays the prediction results of the visible modality detector on visible data, while the second row presents the prediction results of the infrared modality detector on infrared data. The third row provides a detailed comparison of the targets from both modalities within the red box areas. Furthermore, the second column displays the object-width-localization prediction heatmap, the third column presents the object-height-localization prediction heatmap, the fourth column shows the pedestrian-category-classification response heatmap, and the fifth column exhibits the bicycle-category-classification response heatmap. The heatmaps associated with object classification exhibit significant differences, whereas those associated with localization are consistent. This is evident in the comparison diagram presented in the last row.

**Figure 2 entropy-25-01166-f002:**
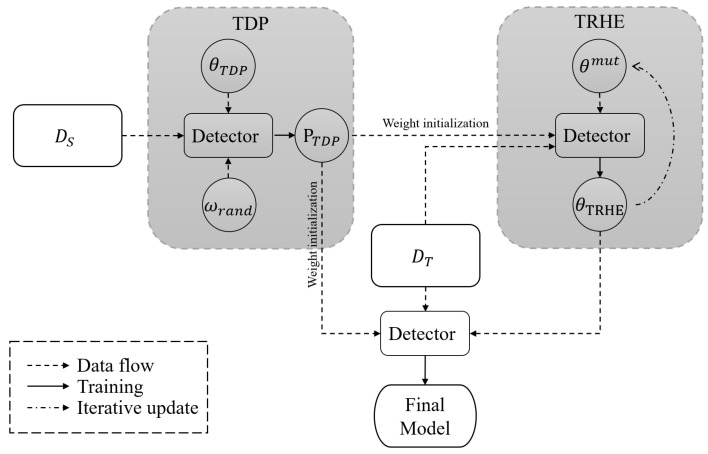
An overview of the algorithms proposed in this paper.

**Figure 3 entropy-25-01166-f003:**
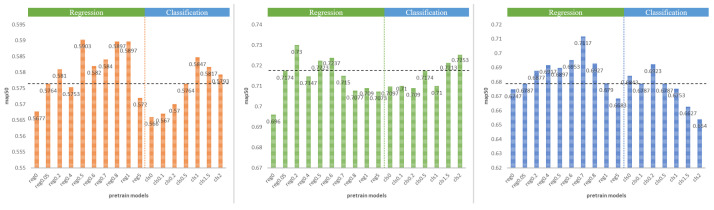
The results of various pre-training models on the MSOD dataset’s multiple modalities, with the black dotted line representing the baseline. The orange table represents the result of the near-infrared modality, the green table represents the result of the medium-wave modality, and the blue table represents the result of the long-wave infrared modality. The $G_{C}$ of reg models is 0.5, and the $G_{R}$ of cls models is 0.05.

**Figure 4 entropy-25-01166-f004:**
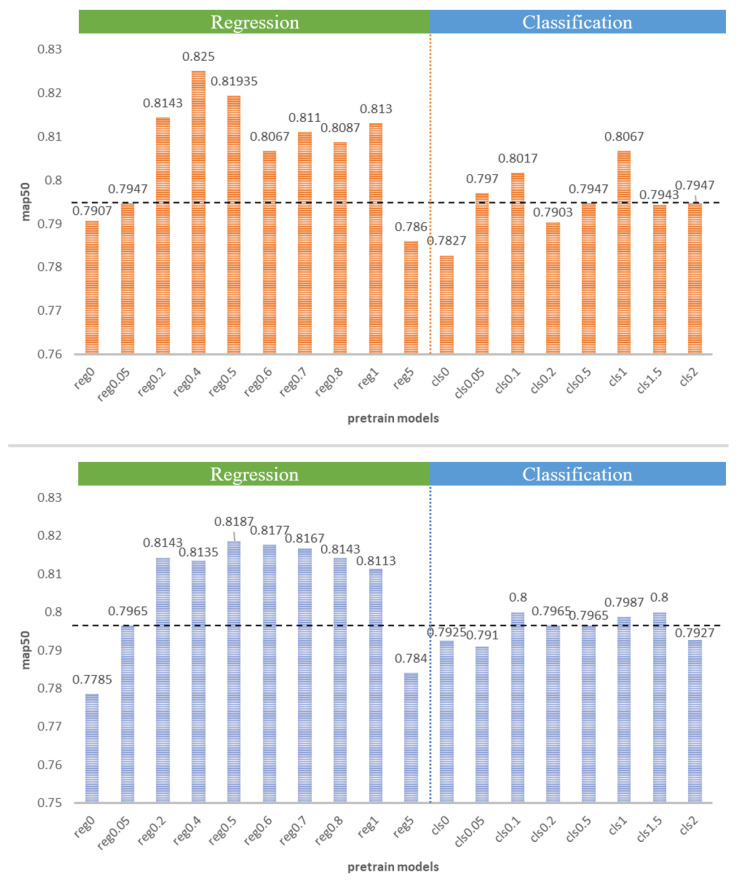
The results of various pre-training models on the FLIR dataset’s infrared and mixed modalities, with the black dotted line representing the baseline. The orange table shows the results on the infrared modality and the blue table shows the results on the mixed modality. The $G_{C}$ of reg models is 0.5, and the $G_{R}$ of the cls models is 0.05.

**Figure 5 entropy-25-01166-f005:**
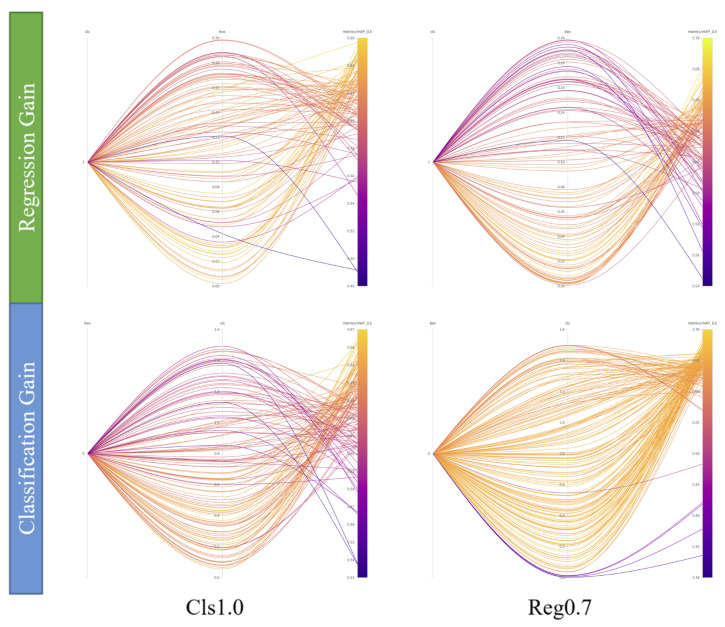
Visualization of guided hyperparameter sweep results for two example pre-training models. The results of the regression and classification attribute hyperparameter sweep based on the Cls1.0 pre-training model are shown in the left column, and the results of the regression and classification attribute hyperparameter sweep based on the Reg0.7 pre-training model are shown in the right column. There are four subplots, and each subplot contains three vertical axes: the cls axis for the value of $G_{C}$, the box axis for the value of $G_{R}$, and the metric axis for the final training accuracy. Each curve connects a point on each of the three vertical axes, corresponding to the values of $G_{C}$ and $G_{R}$ and the resulting mAP accuracy for single experiments.

**Table 1 entropy-25-01166-t001:** Comparison between the proposed method and the multi-spectral and single-spectral state-of-the-art methods; the best result is in bold, and the second best is underlined.

Method	Data	Person	Bicycle	Car	mAP50
MMTOD-UNIT [36]	RGB + Thermal	64.5	49.4	70.8	61.5
CFR [35]	RGB + Thermal	74.49	55.77	84.91	72.39
SSTN101 [37]	RGB + Thermal	-	-	-	77.57
GAFF [38]	RGB + Thermal	-	-	-	72.90
ProbEn [39]	RGB + Thermal	87.65	73.49	90.14	83.76
ThermalDet [23]	Thermal	78.24	60.04	85.52	74.60
$GAN_{b⟶a}$ [8]	Thermal	78.24	60.04	85.52	74.60
Ours(yolom)	Thermal	86.23	71.78	90.5	82.83
Ours(yolox)	Thermal	**86.93**	**75.05**	**91.2**	**84.4**

**Table 2 entropy-25-01166-t002:** The results of fine tuning based on the pre-training model constrained by different regression attributes on the MSOD long-wave infrared dataset.

$G_{C}$	$G_{R}$	mAP50	Person	Car	Bike	Improvement
0.5	0	67.47	83.87	61.5	57.07	−0.4
0.5	0.05	67.87	83.87	62.3	57.43	-
0.5	0.2	68.77	84.17	61.93	60.17	0.9
0.5	0.4	69.17	84.67	63.03	59.73	1.3
0.5	0.5	68.97	84.8	60.97	61.23	1.1
0.5	0.6	69.53	85.53	61.67	60.23	1.66
0.5	0.7	**71.17**	85.43	**66.27**	**61.8**	**3.3**
0.5	0.8	69.27	**86.53**	60.67	60.63	1.4
0.5	1.0	67.9	85.07	62.83	55.93	0.03
0.5	5.0	66.83	84.93	61.57	54	−1.04

**Table 3 entropy-25-01166-t003:** The results of fine tuning based on the pre-training model constrained by different classification attributes on the MSOD long-wave infrared dataset.

$G_{C}$	$G_{R}$	mAP50	Person	Car	Bike	Improvement
0	0.05	68.43	84.07	62.4	58.77	0.56
0.1	0.05	67.87	84.2	61.23	58.23	0
0.2	0.05	**69.23**	**84.6**	**62.33**	**60.8**	**1.36**
0.5	0.05	67.87	83.87	62.3	57.43	-
1.0	0.05	67.53	83.08	61.48	58.08	−0.34
1.5	0.05	66.27	81.33	62.1	55.27	−1.6
2.0	0.05	65.4	80.73	61.77	53.67	−2.47

**Table 4 entropy-25-01166-t004:** Channel-mixing result of visible and infrared modality on the FLIR dataset.

Image Type	mAP50	Person	Bicycle	Car
RGB	67.2	67	52.6	82
IR	78.1	82.4	62.1	89.9
RGT	76.1	80.6	58.9	88.9
RTB	77.5	83.4	59.4	89.6
TGB	76.7	81.8	59.7	88.6
RTT	79.5	**84.1**	63.9	**90.5**
TGT	78.6	82	64.5	89.3
TTB	**81**	83.3	**69.4**	90.2

**Table 5 entropy-25-01166-t005:** The results of fine tuning based on the pre-training model constrained by different regression attributes on the FLIR infrared dataset.

$G_{C}$	$G_{R}$	mAP50	Person	Bicycle	Car	Improvement
0.5	0	79.07	82.4	65.2	89.7	−0.4
0.5	0.05	79.47	83.73	64.63	90.33	-
0.5	0.2	81.43	85.6	68.3	90.33	1.96
0.5	0.4	**82.5**	**85.9**	**71**	90.63	**3.03**
0.5	0.5	81.67	85.5	68.83	**90.77**	2.2
0.5	0.6	80.67	85.3	66.23	90.5	1.2
0.5	0.7	81.1	85.43	67.37	90.47	1.63
0.5	0.8	80.87	84.83	67.3	90.6	1.4
0.5	1.0	81.3	85.53	67.6	**90.77**	1.83
0.5	5.0	78.6	84.87	61.2	89.7	−0.87

**Table 6 entropy-25-01166-t006:** The results of fine tuning based on the pre-training model constrained by different classification attributes on the FLIR infrared dataset.

$G_{C}$	$G_{R}$	mAP50	Person	Bicycle	Car	Improvement
0	0.05	78.27	84.17	60.3	**90.47**	−1.36
0.1	0.05	79.7	**84.5**	64.63	89.97	0.07
0.1	0.05	80.17	83.13	67.47	89.9	0.54
0.2	0.05	79.03	83.47	63.33	90.27	−0.6
0.5	0.05	79.47	83.73	64.63	90.33	-
1.0	0.05	**80.67**	83.67	**68.1**	90.17	**1.04**
1.5	0.05	79.43	83.03	65.13	90.07	−0.2
2.0	0.05	79.47	83	65.5	89.87	−0.16

**Table 7 entropy-25-01166-t007:** Results of task-related hyperparameter evolution experiments.

Method	Initial $G_{C}$	Initial $G_{R}$	Final $G_{C}$	Final $G_{R}$	Parents Select	mAP50	mAP	Fitness
Baseline	0.2	0.02	0.2	0.02	-	**82.4**	42	46.04
TRHE	0.2	0.02	0.22955	0.02101	Random	**82.4**	41.97	46.013
TRHE	0.2	0.02	0.2042	0.02216	Best	81.87	42.5	46.437
TRHE*	0.2	0.02	0.24544	0.02454	Best	81.87	42.5	46.437
Baseline	0.5	0.05	0.5	0.05	-	81.935	42.285	46.25
TRHE	0.5	0.05	0.5	0.05	Random	81.2	42.97	46.793
TRHE	0.5	0.05	0.47716	0.05937	Best	81.97	42.03	46.024
TRHE*	0.5	0.05	0.48879	0.04888	Best	82	**43.27**	**47.143**
Baseline	0.8	0.08	0.8	0.08	-	81.55	42.15	46.09
TRHE	0.8	0.08	0.8	0.08	Random	81.4	42.9	46.75
TRHE	0.8	0.08	0.8	0.08	Best	82.2	42.97	46.893
TRHE*	0.8	0.08	0.8	0.08	Best	81.5	42.5	46.4

**Table 8 entropy-25-01166-t008:** Results of ablation experiments.

Method		MSOD Dataset				FLIR Dataset			
**TDP**	**TRHE**	**mAP50**	**Improvement**	**mAP**	**Fitness**	**mAP50**	**Improvement**	**mAP**	**Fitness**
		67.87	-	37.8	40.807	79.47	-	40.67	44.55
*√*		71.17	3.3	39.67	42.82	82.5	3.03	42.53	46.527
	*√*	69.57	1.7	39.3	42.327	79.67	0.2	42.03	45.794
*√*	*√*	**71.33**	**3.46**	**40.2**	**43.313**	**82.83**	**3.36**	**42.78**	**46.785**

## Data Availability

Not applicable.

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
