# Peer review of "Task-Decoupled Knowledge Transfer for Cross-Modality Object Detection"

_entropy, 2023, doi:10.3390/e25081166_

Round 1

Reviewer 1 Report

This research article presents a novel approach to cross-modality object detection that uses infrared modality as a supplement or replacement for visible modality. The authors investigate the impact of various task-relevant features on cross-modality object detection and suggest a knowledge transfer algorithm based on classification and localization decoupling analysis. The paper proposes a task-decoupled pre-training method to adjust the attributes of various tasks learned by the pre-training model and a task-relevant hyperparameter evolution method to increase the network's adaptability to attribute changes in pre-training weights. The results show that the proposed method improves the accuracy of multiple modalities in multiple datasets and reaches the state-of-the-art level in the FLIR ADAS dataset.

Overall, this is a well-written and informative article that presents a novel approach to cross-modality object detection. The methodology is well-explained, and the results are presented clearly and convincingly. The authors provide sufficient detail on their experiments and analysis, and the conclusion is well-supported by the results. The research presented in this article is valuable and can have a significant impact on the field of cross-modality object detection. Therefore, I would recommend a last English proofreading of the entire article before accepting the manuscript.

Overall the article can be followed. However, a last English proofreading with a native speaker is suggested to increase the quality of the manuscript.

Reviewer 2 Report

1. What is the red box in Figure 1? What is used for? What is the heatmap? How to obtain it?

2. The notations in equations 1 and 2 should be defined in detail.

3. In equation 3, three coefficients were added into the loss function. How to obtain these three coefficients? How to guarantte the loss function can learn more from regression features? Please justify it.

4. Which part is used to decouple classification and localization features? Equation 3? They are still integrated in the loss function.

5. What do the curves mean in Figure 5.

NA
